# Expression Characteristics in Roots, Phloem, Leaves, Flowers and Fruits of Apple circRNA

**DOI:** 10.3390/genes13040712

**Published:** 2022-04-18

**Authors:** Dajiang Wang, Yuan Gao, Simiao Sun, Lianwen Li, Kun Wang

**Affiliations:** Key Laboratory of Horticulture Crops Germplasm Resources Utilization, Research Institute of Pomology, Chinese Academy of Agricultural Sciences (CAAS), Ministry of Agriculture and Rural Affairs of the People’s Republic of China, Xingcheng 125100, China; dajiang0101@126.com (D.W.); gaoyuan02@caas.cn (Y.G.); sunsimiao@caas.cn (S.S.); lilianwen@caas.cn (L.L.)

**Keywords:** apple, circRNA, tissue, spatial expression, qRT-PCR

## Abstract

Circular RNAs (circRNAs) are covalently closed non-coding RNAs that play pivotal roles in various biological processes. However, circRNAs’ roles in different tissues of apple are currently unknown. A total of 6495 unique circRNAs were identified from roots, phloem, leaves, flowers and fruits; 65.99% of them were intergenic circRNAs. Similar to other plants, tissue-specific expression was also observed for apple circRNAs; only 175 (2.69%) circRNAs were prevalently expressed in all five different tissues, while 1256, 1064, 912, 904 and 1080 circRNAs were expressed only in roots, phloem, leaves, flowers and fruit, respectively. The hosting-genes of circRNAs showed significant differences enriched in COG, GO terms or KEGG pathways in five tissues, suggesting the special functions of circRNAs in different tissues. Potential binding interactions between circRNAs and miRNAs were investigated using TargetFinder; 2989 interactions between 647 circRNAs and 192 miRNA were predicated in the present study. It also predicted that Chr00:18744403|18744580-mdm-miR160 might play an important role in the formation of flowers or in regulating the coloration of flowers, Chr10:6857496|6858910–mdm-miR168 might be involved in response to drought stress in roots, and Chr03:1226434|1277176 may absorb mdm-miR482a-3p and play a major role in disease resistance. Two circRNAs were experimentally analyzed by qRT-PCR with divergent primers, the expression levels were consistent with RNA-seq, which indicates that the RNA-seq datasets were reliable.

## 1. Introduction

Circular RNA (circRNA), as the name suggests, is a type of single-stranded non-coding RNA, which is mainly produced by the presence of a covalent bond, linking the 3′ and 5′ ends generated by pre-mRNA through variable shearing processing; it was first identified in a plant virus in the 1970s [1,2,3]. Plant circRNAs were first discovered in *Arabidopsis* in 2014 [4]; more than 15 kinds of monocots and dicots plants have recently been reported regarding circRNA research [5]. The length of circRNAs ranges from <100 nt to larger than >4000 nt [6], but it is commonly a few hundred nucleotides [7,8]. CircRNA is abundantly present in the eukaryotic transcriptome; their important role in life activities has recently gradually being discovered [4,8,9,10].

There are different kinds of circRNA according to their origin in plants, including exon-cricRNA, intron-circRNA, exon-intron-circRNA, intergenic circRNA, antisense-circRNA and so on [9,11,12,13,14]. The functions of circRNA are varied in plants, such as regulating gene expression and tissue development through temporal and spatial expression characteristics, responding to biotic and abiotic stress, participating in regulating the transcription of its source gene, involvement in regulating the formation of miRNA, adsorbing miRNA and then up-regulating the expression of its target genes [9,10,15,16,17,18,19,20,21,22].

Compared with circRNA research in animals, there are few research reports in plants, and even fewer in fruit trees. Pear was reported to have about 33 circRNAs involved in drought stress [23]; circRNA had a specific response to the invasion of pathogens in kiwifruit [21]; and 29 circRNAs were potential targets for 16 miRNAs in trifoliate orange [23]. Apple is one of the most important fruit crops in the world; it plays an important role in agriculture. Recently, several versions of genome sequences have been published [24,25,26,27,28], but there have been few reports on circRNA research on apple up to now. CircRNAs have tissue specificity and regulate a wide range of biological progresses related to biotic/abiotic responses and plant development [17,20,29,30], so this study focused on the difference in circRNA expression in roots, phloem, flowers, leaves and fruits of ‘Gala’ apple using rRNA-depleted total RNA-seq; we hope it may provide basic understanding for the role of circRNA in apple trees.

## 2. Materials and Methods

### 2.1. Experimental Materials

Samples of flowers, leaves, phloem and fruit were collected from ‘Gala’ apple trees, which were planted in the National Repository of Apple Germplasm Resources (Xingcheng, China) in 2002; *Malus baccata* (L.) Borkh. was the rootstock. Samples of leaves and phloem were collected after germination in spring, flower samples were collected when flowers were in the balloon period, and fruit samples were collected at physiological maturation. Root samples were harvested from the tissue culture plantlets. All samples were frozen by liquid nitrogen and stored at −80 °C in fridge. Each line had three biological repetitions.

### 2.2. Genome-Wide circRNAs Sequencing

Total RNA was extracted and rRNA was removed using the Epicenter Ribo-Zero^TM^ (Epicentre, Madison, WI, USA) following the manufacturer’s procedure. The total RNA quantity and purity were analyzed using Aglient 2100 bioanalyzer test (the RIN value of total RNA was ≥7.0, 28S/18S ≥ 1.0) (Agilent, Santa Clara, CA, USA), Qubit 2.0 detection (total RNA concentration ≥ 65 ng/uL) and Nanodrop detection (OD260/280 ≥ 1.8, OD260/230 ≥ 0.5).

Epicentre epicentre Ribo-Zero^TM^ kit was used to remove rRNA; rRNA-depleted RNA was interrupted randomly by Fragmentation Buffer; random hexamer primer was used to synthesize, first, cDNA; then, second, cDNA was synthesized; cDNA was purified by AMPure XP beads; the purified double-stranded cDNA was repaired; A was added and sequenced; AMPure XP Beads were used for fragment size selection; the U chain was degraded; and then, the cDNA library was enriched by PCR. After the construction of the library, the concentration and insert size of the library was detected using Qubit 2.0 and Agilent 2100, respectively. The effective concentration of the library was accurately quantified by the Q-PCR method so as to ensure the library quality (determined to be >2 nM). After detection, different libraries were pooled based on the target machine data volume, and were sequenced on the Illumina Hi-Seq platform of BioMarker Technologies (Beijing, China).

### 2.3. Identification and Differential Expression Profile of circRNAs in Apple Tissues

We predicted the circRNAs using the find_circ software. First, the find_circ software took 20 bp as the anchor point at both ends of the reads on the genomic alignment; then, it compared the anchor points as independent reads to the ‘Golden delicious’ apple GDDH13 reference genome, and found the only matching site where the alignment positions of the two anchors were reversed in a linear direction. Afterwards, the reads of the anchor would be extended until the junction position of the circular RNA were found. If the sequences on both sides were GT/AG splicing signals, respectively, they would be determined as being circRNA.

### 2.4. Functional Annotation Analysis of Parental Genes of circRNAs

Functional annotation analysis was performed to evaluate the potential functions of the parental genes of circRNAS. GO (Gene Ontology Consortium) categories, COG (Cluster of Orthologous Groups of proteins) categories and KEGG (Kyoto Encyclopedia of Gene and Genomes) annotation were carried out to analysis the function of parental genes of circRNAs; then, the function of circRNAs was predicted.

### 2.5. Prediction of miRNA Target Sites in circRNAs

We identified targets of miRNAs using TargetFinder, based on the known miRNAs, the newly predicted miRNAs, and the gene sequence information in ‘Golden delicious’. As circRNAs contain multiple miRNA binding sites, the miRNA target gene prediction methods can be used to identify the circRNAs that bind to miRNAs, and the functions of the circRNAs can be elucidated based on the functional annotation of the miRNA target genes.

### 2.6. Quantitative Real-Time PCR Validation

Quantitative real-time PCR (qRT-PCR) was carried out to validate differential expressional levels of circRNAs from ‘Gala’ apple. According to the instruction of the TUREscript 1st Stand cDNA SYNTHESIS Kit (Aidlab, China), 800 ng of the total RNA was reverse-transcribed with random primers. The reaction system included 800 ng RNA, 4 μL 55× RT Reaction Mix, 0.5 μL Rondam primer/oligodT, 0.5 μL N6, 0.8 μL TUREscript H^−^ RTase/RI Mix, added RNase Free dH_2_O to 20 μL; reaction conditions were 42 °C 40 min, and 65 °C 10 min.

Divergent primers were designed using an ‘out-facing’ strategy in order to obtain amplicon from circle template (Appendix A). Quantitative real-time PCR (qRT-PCR) was performed using SYBR Green Master Mix (Vazyme, Nanjing, China). The qRT-PCR aliquot contained 1 μL cDNA, 3 μL ddH_2_O, 0.5 μL of each forward and reverse primer (200 nM), 5 μL 2× SYBR^®^ Green Supermix. The reaction conditions included initial denaturation at 95 °C for 3 min, followed by 39 cycles at 95 °C for 10 s, 60 °C for 30 s, melt curve analysis (60~95 °C, +1 °C/cycle, holding time 4 s). The levels of circRNAs were normalized by citrus *β*-Actin. All real-time PCR assays were performed with three biological replicates. The relative expression levels were calculated with the 2^−ΔΔCt^ method.

## 3. Results

### 3.1. Identification and Characteristics of circRNAs in ‘Gala’ Apple

The circRNAs were determined at five different tissues of ‘Gala’ apple. For each biological, more than 110 million unmapped reads from roots, phloem, leaves, flowers and fruit were produced for circRNA identification, respectively. After mapping to the apple reference genome, 147,848,711; 142,596,329; 145,428,347; 127,673,515 and 146,376,167 raw reads corresponding to 127,226,811; 145,235,124; 143,705,652; 141,886,898 and 146,022,255 mapped reads were identified in the roots, phloem, leaves, flowers and fruit according to the circRNAs sequencing (Appendix A). On average, the ratio of mapped reads, and Q30 and GC content in the five libraries were 99.2%, 94.5% and 46.0%, respectively (Appendix A).

In total, 6495 circRNAs were identified at five tissues of ‘Gala’ apple, and 5163 circRNAs were distributed in the 17 chromosomes. It revealed a non-random distribution of circRNAs in the chromosomes. Some chromosomal regions lacked circRNAs, and some regions had a high density of circRNAs (Figure 1). In total, 517 and 272 circRNAs were from chromosome 15 and chromosome 14, which accounted for the most and the least (8.0% and 4.2%, respectively); the numbers of circRNAs in other chromosomes ranged from 287 to 483 (Appendix A).

We found most of the circRNAs identified were <1400 nt in length (73.9% of total circRNAs); the exon circRNA predominated in short circRNAs, some were very long (1130 circRNAs > 10 kb), but nearly all of them were intergenic circRNAs; the proportion of intergenic circRNAs increased as the length increased (Appendix A; Figure 2a). This might be partially caused by the imperfect assembly of the apple reference genome sequences, which were >58% repetitive DNA [26,27,28].

Among the 6495 circRNAs, circRNAs from exon, intron and intergenic regions accounted for 65.99%, 28.47% and 5.54%, respectively (Figure 2b). Through comparison with mRNA, we found that 4646 circRNAs were derived from 3545 unique hosting protein-coding genes, and 34 of which were new genes. About 19.52% (692) of hosting protein-coding genes could generate more than one circRNA isoform; for example, MD17G1110200 could generate 27 circRNAs (Appendix A).

Back-splicing sites play a decisive role in the origination of circRNA. The formation of circularization or the lack of circularization was determined only by the back-splicing site. Therefore, structures of circRNAs were analyzed based on their host genes. Among 6495 circRNAs, a total of 2157 circRNAs originated from 407 host-genes and 335 intergenic regions; alternative back-splicing circularization events were also identified (Appendix A).

### 3.2. Different Expression in Tissues of ‘Gala’ Apple

To evaluate the consistency of the experimental materials and the accuracy of sequencing, we carried out the Pearson correlation coefficient of three biological replicates on roots, phloem, leaves, flowers and fruits; gene expression levels and the number of overlapping circRNAs in each biological replicate were relatively consistent (Appendix A).

The prevalent and spatial expression of circRNAs in ‘Gala’ apple were studied. The prevalence was relatively low in different tissues; in total, 175 circRNAs were shared in the five libraries, while 1256, 1064, 912, 904 and 1080 circRNAs expressed specifically in roots, phloem, leaves, flowers and fruit, respectively (Figure 3a; Appendix A). We normalized expression profiles of 175 prevalent circRNAs based on their TPM values, which permitted quantitative comparisons of the levels of each circRNA among the different tissues. The relative expression of circRNAs from fruits was down-regulated compared with other tissues (Figure 3b). Some circRNAs were highly enriched in flowers; for example, the expression of Chr03:36631963|36634436 was 5.4 times of that of leaves, while Chr05:4103761|4104135 was enriched in phloem and roots; Chr10:20065036|20065186 was enriched in leaves, while Chr15:21425004|21425163 was enriched in fruit, which was also enriched in leaves and flowers; its host-gene was cold-regulated 47, indicating this circRNA may be a response to cold in fruits, leaves and flowers; these tissues were sensitive to cold. These results indicated that expression of circRNAs had tissue specificity.

### 3.3. Functional Annotation of Host-Genes of circRNAs

To investigate the putative functions of host-genes of circRNAs, COG (1537), GO (1431), KOG (2265), Swissprot (2730), eggNOG (3414) and KEGG (1559), enrichment analyses were carried out; 3496 hosting-coding genes were annotated in all (Appendix A). We observed that the hosting protein-coding genes of circRNAs in GO were significantly enriched in biological processes related to the metabolic process, cellular process, single-organism process, response to stimulus and so on; in the cellular components related to the cell part, cell, organelle, membrane, etc.; and in the molecular function related to binding, catalytic activity, transporter activity, etc. Moreover, COG terms, such as general function prediction, transcription, replication, recombination and repair, signal transduction mechanisms and posttranslational modification, protein turnover and chaperones, were also significantly enriched. The KEGG pathway enrichment analysis showed that the hosting protein-coding genes were enriched in the biosynthesis of amino acids, protein processing in endoplasmic reticulum, and ribosome and plant hormone signal transduction in the top four numbers of genes involved (Figure 4 and Appendix A).

Considering the tissue samples we used in the current study, these circRNAs and their parental genes may play familiar or different roles in different tissues. For host-genes of the 175 prevalent circRNAs, COG terms were mainly enriched in posttranslational modification, protein turnover, chaperones, transcription and replication, and recombination and repair. For specific host-genes of circRNAs in different tissues, their main function in COG terms was shown in general function prediction only, transcription, replication, recombination and repair, posttranslational modification, protein turnover, chaperones and so on. However, there were greater numbers of intracellular trafficking, secretion, vesicular transport, energy production and conversion than others; general function prediction, transcription, posttranslational modification, protein turnover and chaperones mainly were enriched in flowers (Figure 5).

### 3.4. CircRNA–miRNA Interaction Network in Apple

Recent studies have reported that circRNAs can act as endogenous target mimics (eTMs) for certain miRNAs and prevent miRNAs from regulating their target genes. To uncover whether circRNAs can target miRNAs and further affect the post-transcriptional regulation of genes in apple, potential binding interactions between circRNAs and miRNAs were investigated. In total, 647 circRNAs (9.96%) contained at least one predicted binding site for 192 miRNAs when searched across the miRNAs of *Malus* in the miRbase (Appendix A). Such well-known miRNAs as mdm-miR156, mdm-miR172 and mdm-miR395 were predicted to be targeted by specific circRNAs in different apple tissues. Multiple circRNAs (or miRNAs) were predicted to interact with more than one miRNA (or circRNA). Some key circRNAs or miRNAs that were thought to play key roles in the circRNA-miRNA interaction were accentuated in the network. For example, 1026 interactions were predicted between circRNAs and the mdm-miR156 family, while 564 interactions could be predicted between circRNAs and the mdm-miR172 family. Meanwhile, 38 miRNAs were predicted to bind to Chr07:1752264|1819061 and Chr07:1752869|1819617, while 32 miRNAs can be targeted by Chr11:2667180|2760235 (Appendix A).

Furthermore, miRNAs also have the characteristics of spatial-temporal expression; interactions between circRNAs and miRNAs were featured to illustrate the function in different tissues. There were 163 interactions in all five tissues between 18 miRNA families and 38 circRNAs, including mdm-miR1511, mdm-miR156, mdm-miR162, and mdmmiR164 and so on; most interactions were with circRNAs and mdm-miR156. There were 431, 414, 214, 290, and 301 interactions only in roots, phloem, leaves, flowers and fruits, respectively (Appendix A). Possible specific and common miRNAs were analyzed in five tissues; the Venn diagram showed that there were 117 miRNAs that could be targets in all tissues, 3 miRNAs in phloem and 5 miRNAs in flowers and roots (Figure 6a; Appendix A). In particular, mdm-miR160 families only had interactions in flowers, mdm-miR168 families and mdm-miR394 families in roots; they may have special expression and regulation characteristics among different tissues.

The network of circRNA-miRNA-mRNA was analyzed; we analyzed the correlation of expression in five tissues between circRNAs and mRNAs. The expression of circRNA and mRNA targeted by miRNA should be positively correlated. We extracted circRNAs and mRNAs whose correlation was greater than 0.8 at the significant level of 0.05, then a regulatory network diagram was constructed (Figure 6b; Appendix A); the possible role of circRNAs was speculated through the functional annotation of mRNA. As Figure 6b and Appendix A showed, one circRNA can be a ceRNA for different miRNA in apple; also, the same miRNA can be absorbed by different circRNAs. The regulatory network included functional genes, transcription factors, etc. It was interesting that the garget genes of mdm-miR482a-3p were mostly disease-resistant genes; this indicates that Chr03:1226434|1277176 may play a major role in disease resistance for apple.

### 3.5. qRT-PCR Validation

To confirm our identification of circRNAs, two random-selected circRNA candidates were selected for experimental validation using quantitative reverse transcription PCR (qRT-PCR). Divergent primers were designed from both side sequences of candidate circRNAs. As shown in Figure 7, in this study, two random-selected circRNAs showed consistent expression patterns with RNA-seq results; the R square between RNA-seq and qRT-PCR were 0.8678 and 0.8821 for Chr11:1679797|1735868 and Chr01:21162287|21214009, respectively. The results of this experiment indicated that the circRNAs identified in the RNA-seq datasets were reliable.

## 4. Discussion

As a rising star of non-coding RNAs, circRNAs widely exist in plants and are involved in many biological processes [31]. So far, several versions of the genomic information of Golden delicious have been reported [24,25,26], and other cultivars have also had their genomes published [27,28]; however, studies on circRNAs in important tissues of apple are still lacking. Here, we reported a genome-wide identification and potential functional analysis of circRNAs in roots, phloem, leaves, flowers and fruits of ‘Gala’ apple, which is a top cultivar grown worldwide and an important parent for apple breeding. We obtained 318.50 Gb of clean data; the percentage of Q30 was 91.99% on average, the ratio of mapped reads was 96.39~99.89%, and the R square of expression by RNA-seq and qRT-PCR was more than 0.8; all above indicated the RNA-seq datasets were reliable.

A total of 6495 circRNAs were identified from five tissues of ‘Gala’ apple; the length of 73.9% of circRNAs was >1400 nt, which was consistent with pear, orange and other species [21,23,32]; the greatest distribution of circRNAs was in chromosome 15, and the least was in chromosome 14, indicating that the distribution of circRNAs was somewhat favorable. Earlier studies revealed that circRNAs exhibited developmental-specific expression patterns in plants [17]. Here, we compared the expression of circRNAs in different tissues of ‘Gala’ apple; the expression of circRNAs showed obvious spatial characteristics, where the number of circRNAs identified in different tissues was distinguishing, with the greatest being in roots and the least in flowers. In differential expression analysis, the greatest distribution of circRNAs from five tissues was in chromosome 2, which did not match with the finding that the distribution of circRNAs identified in chromosome 15 was the greatest. This indicated that the different expression of circRNAs identified from chromosome 2 was the expression quantity, whereas chromosome 15 showed organizational difference.

CircRNAs are usually generated from exonic, intergenic and intronic regions in various species, but the main kind of plant circRNAs is exon circRNAs [6,17,33,34]. In the present data, the proportion of exon circRNAs was 65.99%, which was consistent with most circRNAs mainly originating from the transcription regions of both monocot and dicot plants [17]. There are two different alternative circularization patterns, namely, alternative back-splicing circularization (the same host gene, different back-splices) and alternative splicing circularization (the same back-splice, different circularization structure). A total of 2157 circRNAs originated from alternative circularization, which provided reliable evidence that apple circRNAs had a great number of alternative circularization isoforms, which is consistent with previous reports of other plant species [6,17,35].

It has been reported that circRNAs could regulate gene expression at posttranscriptional levels; they could adsorb miRNAs and then up-regulate the expression of their target genes [15,36]. Here, we predicted the target miRNAs of circRNAs, and a total of 192 miRNA families had target sites on 647 circRNAs; most of them were intergenic circRNAs, which totaled 89.03% in all. Among the 647 circRNAs, 70 were from chromosome 2, which was the most, and 4 were from chromosome 13, which was the least. There were 3080 interactions between miRNAs and circRNAs; one circRNA can target multiple miRNAs, and the same miRNA can be targeted by multiple circRNAs; this was consistent with other reports in plants [27,37].

Both circRNAs and miRNA have spatial-temporal expression characteristics [6,21,38]; a network of circRNAs-miRNAs-mRNAs was constructed, and it was found that Chr03:1226434|1277176 may absorb mdm-miR482a-3p and play a major role in disease resistance in five tissues of apple. We also compared the interactions of miRNA and circRNA in different tissues. This showed that tissue specific interactions, such as between mdm-miR160 families and circRNAs, were only obtained in flowers; miR160 could regulate auxin-mediated development by post-transcriptional regulation of the auxin response factors ARF10/16/17, then regulate leaf and flower development [39]. Mdm-miR160 could participate in root formation of apple rootstock and regulate leaf coloration in *Malus* spp. [40,41]. Here, interactions between circRNAs and mdm-miR160 were only identified in flowers, rather than in other tissues; this indicates that Chr00:18744403|18744580 can participate in the formation of flowers or regulate the coloration of flowers.

Mdm-miR168 and mdm-miR394 families that interacted with circRNAs were only obtained in roots. These two miRNAs were mainly a response to stress in plants, and miR168 was only expressed in certain tissues of plants; it could, though, be detected in the roots of soybean [42]. CircRNAs, which were predicted to interact with mdm-miR168 and mdm-miR394, were Chr10:6857496|6858910 and Chr10:41060498|41137024, respectively, and the host gene of Chr10:6857496|6858910 was MD10G1051700, which was described as being calmodulin-binding transcription activator 2-like; its functions include regulation of growth and development, defense response, general stress response, frost resistance, drought resistance, hormones signaling pathways and so on [43,44,45,46,47,48,49,50,51,52]. Other reports have shown that the miR168 of roots plays an important role in drought stress in plants [53,54,55,56]. Therefore, we hypothesize that Chr10:6857496|6858910 could regulate the expression in response to drought stress in apple roots; there should perhaps be a new focus on the study of apple response to drought stress.

## 5. Conclusions

We identified circRNAs in five tissues of apple ‘Gala’, a variety widely cultivated worldwide. A total of 6495 unique circRNAs were identified from leaves, phloem, flower, fruits and roots of ‘Gala’. In total, 647 circRNAs (9.96%) contained at least one predicted binding site for 192 miRNAs when searched across the miRNAs of *Malus* in miRbase. It predicted that Chr03:1226434|1277176 may absorb mdm-miR482a-3p and play a major role in disease resistance for apple, the network of Chr00:18744403|18744580-mdm-miR160 may play an important role in the formation of flowers or regulating the coloration of flowers, and also that Chr10:6857496|6858910-mdm-miR168 may be involved in response to drought stress in roots.

## Figures and Tables

**Figure 1 genes-13-00712-f001:**
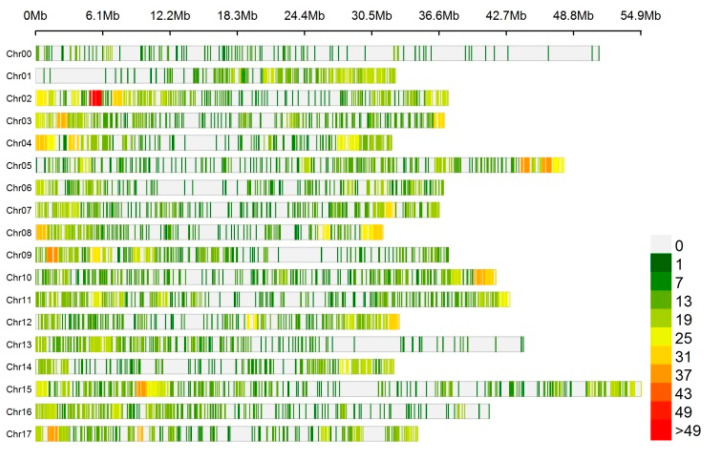
Distribution of circRNAs identified in chromosomes.

**Figure 2 genes-13-00712-f002:**
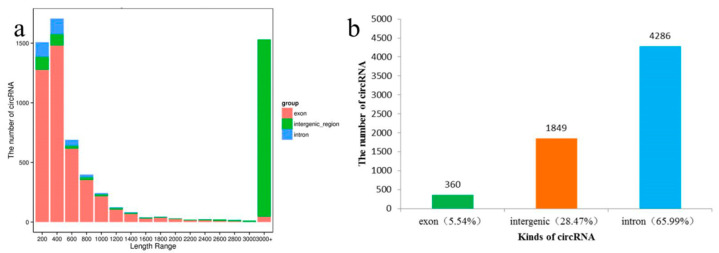
Screening and classification of circRNAs: (**a**) Length distribution of circRNAs identified; (**b**) Number of different kinds of circRANs identified.

**Figure 3 genes-13-00712-f003:**
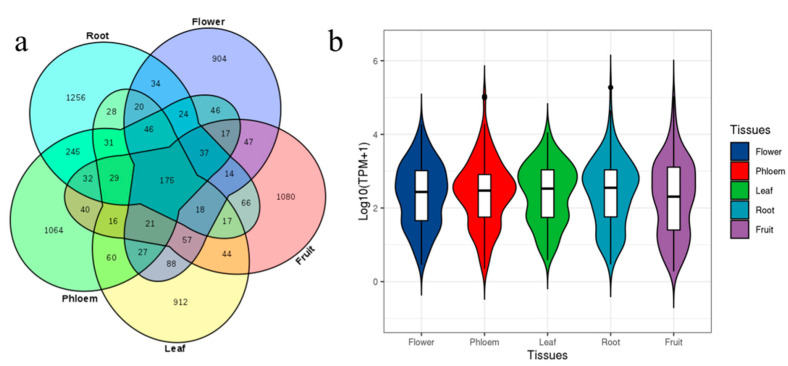
circRNAs in all five tissues and their expression characteristics: (**a**) Venn diagram showing the number of the identified circRNAs in five tissues. (**b**) The overall abundance patterns of parental genes of circRNAs using TPM calculation.

**Figure 4 genes-13-00712-f004:**
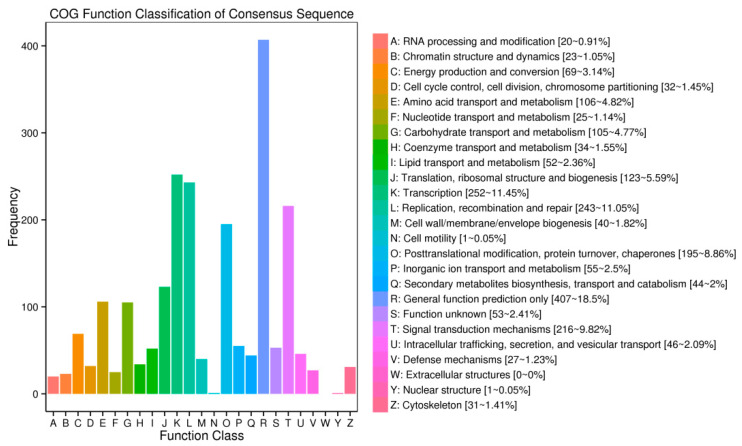
Cluster of genes (COG) classifications of host-genes (protein-based genes) of circRNAs.

**Figure 5 genes-13-00712-f005:**
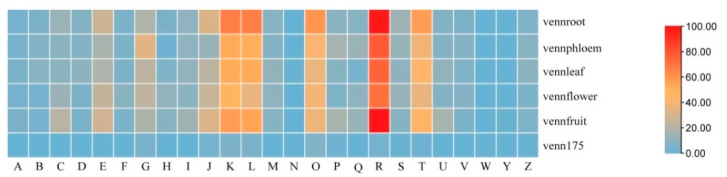
Cluster of genes (COG) classifications of parent genes (protein-based genes) of tissue specific and prevalence expression of circRNAs. (The capital letters represent the function classification of COG, as Figure 4; venn175 represented prevalence expression in all five tissues, vennroot, vennphloem, vennleaf, vennflower and vennfruit representing only the expression in root, phloem, leaf, flower and fruit, respectively.)

**Figure 6 genes-13-00712-f006:**
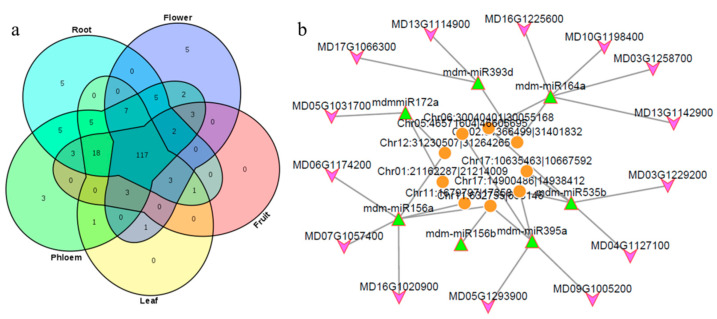
Tissue specific characteristics and function prediction of circRNAs-miRNAs-mRNA: (**a**) Number of the miRNA predicated which had target sites in circRNA identified in five tissues. (**b**) Network predicted of circRNAs-miRNAs-mRNAs in apple (orange circle: circRNAs; green triangle: mdm-miRNAs; purple arrow: mRNAs).

**Figure 7 genes-13-00712-f007:**
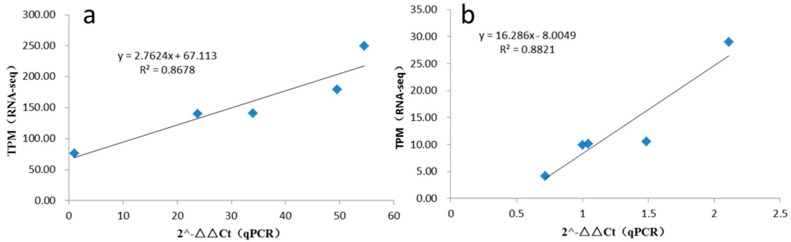
The linear relationship between RNA-seq and qRT-PCR: (**a**) It was qRT-PCR for Chr11:1679797|1735868. (**b**) It was qRT-PCR for Chr01:21162287|21214009.

## Data Availability

The raw sequence data reported in this paper have been deposited in the Genome Sequence Archive (Genomics, Proteomics & Bioinformatics 2021) in National Genomics Data Center (Nucleic Acids Res 2022), China National Center for Bioinformation/Beijing Institute of Genomics, Chinese Academy of Sciences (GSA: CRA006669, accessed on 15 March 2022) that are publicly accessible at https://ngdc.cncb.ac.cn/gsa.

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
