# Peer review of "Expression Characteristics in Roots, Phloem, Leaves, Flowers and Fruits of Apple circRNA"

_genes, 2022, doi:10.3390/genes13040712_

Round 1

Reviewer 1 Report

The English needs improvement before publication.

The work is a solid foundational study of the circular RNAs expressed in apple trees in several tissues. Although there is no direct functional analysis of these RNAs, it is of interest to readers to have them identified and catalogued, and to have a beginning look at the differential expression of the circRNAs over the plant's development.

Author Response

Dear Editors:

We submit our manuscript entitled “Expression characteristics in roots, phloem, leaves, flowers and fruits of apple CircRNA” to “Genes” for publication.

Thank you very much for your valuable suggestions on the manuscript. We have revised the full manuscript according to your suggestions, and added the function analysis. If there are still any inappropriate or incorrect points, please give us suggestions and we will revise it again.

Thank you very much for your attention and consideration.

Best Regards.

Sincerely yours,

Dajiang Wang

Reviewer 2 Report

I did some minor modification as specified in the attached pdf

Author Response

(The authors gave the same response as above.)
